# Porous functionalized polymers enable generating and transporting hyperpolarized mixtures of metabolites

Théo El Daraï[1,2], Samuel F. Cousin[1✉], Quentin Stern[1], Morgan Ceillier[1], James Kempf[3], Dmitry Eshchenko[4], Roberto Melzi[5], Marc Schnell[4], Laurent Gremillard [6], Aurélien Bornet[1], Jonas Milani[1], Basile Vuichoud[1], Olivier Cala[1], Damien Montarnal [2✉] & Sami Jannin [1]

Hyperpolarization by dissolution dynamic nuclear polarization (dDNP) has enabled promising applications in spectroscopy and imaging, but remains poorly widespread due to experimental complexity. Broad democratization of dDNP could be realized by remote preparation and distribution of hyperpolarized samples from dedicated facilities. Here we show the synthesis of hyperpolarizing polymers (HYPOPs) that can generate radical- and contaminant-free hyperpolarized samples within minutes with lifetimes exceeding hours in the solid state. HYPOPs feature tunable macroporous porosity, with porous volumes up to 80% and concentration of nitroxide radicals grafted in the bulk matrix up to 285 μmol g$^{-1}$. Analytes can be efficiently impregnated as aqueous/alcoholic solutions and hyperpolarized up to $P(^{13}C) = 25\%$ within 8 min, through the combination of $^1H$ spin diffusion and $^1H \rightarrow ^{13}C$ cross polarization. Solutions of $^{13}C$-analytes of biological interest hyperpolarized in HYPOPs display a very long solid-state $^{13}C$ relaxation times of 5.7 h at 3.8 K, thus prefiguring transportation over long distances.

[1] Université de Lyon, Centre de RMN à Très Hauts Champs de Lyon, UMR5082 - CNRS/UCBL/ENS de Lyon, Villeurbanne, France. [2] Université de Lyon, CPE Lyon, CNRS, Catalyse, Chimie, Polymères et Procédés, UMR 5265, Lyon, France. [3] Bruker Biospin, Billerica, MA, USA. [4] Bruker Biospin, Fallanden, Switzerland. [5] Bruker Italia Srl, Milano, Italy. [6] Université de Lyon, INSA Lyon, MATEIS UMR CNRS 5510, Bât. Blaise Pascal, Villeurbanne, France. ✉email: samuel.cousin@univ-amu.fr; damien.montarnal@univ-lyon1.fr

Nuclear magnetic resonance (NMR) and magnetic resonance imaging (MRI) have become, within 50 years, techniques of reference for both analytical chemistry and medical diagnostic. The development of very high magnetic fields has led to an incessant increase in resolution and sensitivity, which has enabled faster acquisitions at ever decreasing concentrations. However, sensitivity still remains the Achilles heel of magnetic resonance. Today, the development of hyperpolarization methods to further enhance sensitivity by large factors is one of the most important topics of research in magnetic resonance.

Dissolution dynamic nuclear polarization (dDNP) is one of these hyperpolarization game-changing approaches applicable for both NMR and MRI. dDNP enables hyperpolarization of small molecules therefore spectacularly amplifying their magnetic resonance signals by up to four orders of magnitude[1–3]. Such hyperpolarization is typically carried out in a dedicated apparatus (a dDNP polarizer) and under harsh conditions (high magnetic field, cryogenic temperatures) where the close to unity electron-spin alignment of polarizing agents (PAs, typically molecules containing unpaired electrons) is transferred to the nuclear spins ($^1$H, $^{13}$C, $^{31}$P, $^{15}$N, etc.) of target molecules. Despite promising proofs of concepts such as the detection of intermediates in fast chemical reactions[4], the observation of protein folding in real time[5], or the early detection and monitoring of tumours in humans[6], dDNP remains unfortunately restricted to a small number of specialized research groups around the globe. Indeed, some severe limitations restrain the widespread use of the method, amongst which the experimental complexity involving sterility (for clinical studies[7]), cryogenic temperatures ($1.2 < T < 4.2$ K), high magnetic fields (up to $B_0 = 9$ T[8]), and microwave irradiation sometimes coupled to synchronized radiofrequency irradiations[9,10]. It also requires trained personnel, and finally represents an overall excessive price for the equipment and cryogenic fluids.

Most of these issues would be virtually fixed from the user perspective if one could transport hyperpolarized molecules from a large-scale hyperpolarization preparation centre to the end users' remote locations. In a word, one would need to turn hyperpolarization into a long-lived transportable *consumable*, thus alleviating the struggle of local preparation. Unfortunately, the short-lived character of hyperpolarization in solution (on the order of a minute apart from some exotic cases[11,12] and long-lived states[13–16]), requires that the complex dDNP preparation of the hyperpolarized samples be performed "on-site", next to the NMR or MRI machine.

This fundamental drawback arises from the very nature of dDNP sample formulations featuring an intimate contact of a few nanometres distance between target molecules (typically $^{13}$C-labelled metabolites) and paramagnetic PAs (which was for a long time thought to be necessary for efficient hyperpolarization). Therefore, the PAs and the target molecule are usually homogeneously mixed then shock-frozen in a glassy state. Such rapid freezing ensures a statistical distribution of both components[17,18]. Yet, this close proximity also leads to paramagnetic nuclear spin relaxation induced by electron spin flip-flops of the PAs[19–22], which causes complete hyperpolarization loss within milliseconds in solids[23] to seconds in liquids when the sample is removed from magnetic field of the polarizer and its cryogenic environment.

One promising answer to this roadblock relies on the use of non-persistent PAs such as photoexcited triplets[24] or photo-generated radicals[25–27]. We have recently introduced another approach that deliberately avoids intimate mixing of the target molecules and PAs through phase separation. In this previous work[28], we designed a dDNP sample formulation in which the $^{13}$C-labelled target molecules were provided in the form of micro-crystals (typically 1–10 μm large) dispersed in an organic phase

(for example, toluene/THF) containing the PAs. While direct hyperpolarization of $^{13}$C may appear impossible at first sight in such sample formulations given the long distances (several μm) between $^{13}$C nuclear spins and PAs, the $^1$H abundance in the two phases enables an indirect three-step polarization procedure:

i. Microwave irradiation rapidly builds $^1$H polarization in the frozen solvent containing the PAs.
ii. This $^1$H spin polarization spontaneously diffuses to the $^1$H spins within the micro-crystals.
iii. The $^1$H spin polarization is then transferred to the $^{13}$C spins of the target molecule by irreversible $^1$H $\rightarrow$ $^{13}$C cross-polarization[29,30] (CP).

This method offered hyperpolarization lifetimes exceeding tens of hours on some $^{13}$C-labelled target molecules, thus enabling the transport and storage of hyperpolarized molecules at 4.2 K. This method is, however, restricted to small molecules amenable to well-tailored micro-crystals and involves poisonous organic solvents along with PA residues, which precludes its use with proteins, living cells or animals.

A definitive answer to this issue would be to find a way to generalize the phase-separation strategy to other sample formulations (molecules, mixtures, biological fluids, etc.) without the use of any contaminants (such as the organic solvents hosting the PAs). This would have the potential to transform and democratize the benefits of hyperpolarization to a very wide NMR and MRI community. Target molecules could be hyperpolarized in dedicated centres and delivered as "consumables" ready for dissolution and injection to NMR or MRI.

Many materials containing PAs freely dispersed[31] or covalently bound onto polymers[32,33] and covalent organic frameworks[34] have been designed for MAS- or Overhauser-DNP (i.e. not requiring effective separation of polarizing medium and hyperpolarized analytes). The design of polarizing media for dDNP is, however, more complex as it requires convenient loading and extraction of the analytes as well as optimized separation from the PAs. Previous generations of polarizing materials for dDNP involve, for instance, surface grafted nitroxide mesoporous silica, HYPSO[35–37], that featured quick and efficient $^{13}$C polarization build-up, and also fast relaxation in the solid state (typically $T_1 = 20$ min at 4.2 K) due to short separation length between target molecules and nitroxide radicals (ca. 6 nm). Contaminant-free hyperpolarized solutions can thus be obtained, but off-site transportation could not be envisioned. Other ingenious strategies to control the separation length between analytes and PA involve thermoresponsive polymers[38] or hydrogels[39,40] in which phase separation and collapse of the polymer chains happens upon dissolution with hot water and expels the hyperpolarized analytes and enables separation from the polarizing polymers, but only in the liquid state. In this case also, $^{13}$C relaxation times in the solid state still remain relatively short ($T_1 = 2800$ s) even at low temperatures (1.1 K)[39], which essentially precludes transportation.

Here, we describe the straightforward synthesis of macroporous hyperpolarizing polymers (HYPOP) featuring characteristic separation length above 100 nm, considerably larger than for any previous polarization media, and that contain PAs located not only at the porous surface, but also in the bulk of the polymer. These materials enable efficient $^{13}$C polarization of impregnated metabolite solutions within tens of minutes, and preserve the polarization for hours in the solid state.

## Results and discussion
### Short summary of HYPOP features. 
Figure 1 introduces the various features of HYPOPs, discussed below in this manuscript,

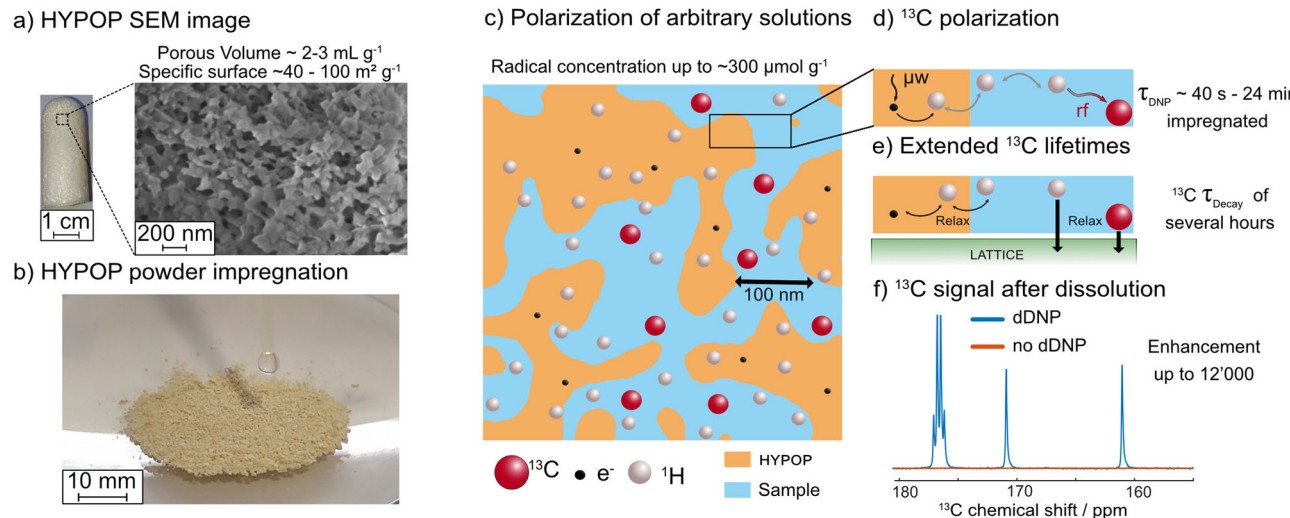

**Fig. 1 Illustration of the steps from production to extraction of transportable hyperpolarization. a** Scanning electron microscopy of the HYPOP material used in this study. **b** Photograph of the impregnation step of powdered HYPOP. **c** Schematic representation of the porous polymer (yellow) with its PAs (black dots) impregnated with a $^{13}$C-labelled molecule (red dots) in aqueous solution (blue). $^{1}$H spins (grey dots) are abundant both in the HYPOP material and aqueous solution. **d** Schematic representation of the polarization transfer from the electrons of the PAs located within the HYPOP material to the $^{1}$H nuclei located within the HYPOP material, followed by $^{1}$H ↔ $^{1}$H spin diffusion across the material interface towards the aqueous frozen solution impregnated in the pores, and finally ended by a cross-polarization (CP) transferring the $^{1}$H polarization to the $^{13}$C spins of the target molecules. **e** Schematic representation of the slow $^{13}$C nuclear spin-lattice relaxation in solid state, mostly free from any paramagnetic relaxation since the $^{13}$C spins are physically well isolated from the PAs. **f** Carbonyl region of the $^{13}$C-NMR spectrum measured on a 80 MHz Bruker BioSpin Fourier 80 benchtop spectrometer (1 scan, 2 s), of a sample containing 1 M [1-$^{13}$C]sodium acetate, 1 M [1-$^{13}$C]sodium formate and 1 M [1-$^{13}$C]glycine, hyperpolarized with HYPOP materials, displaying a liquid-state polarization enhancement exceeding 5000.

that have been tailored with suitable structure and pore size range to give high performances both as polarization source and storage matrix for dDNP applications. HYPOPs are able to withstand both extreme cryogenic temperatures (down to superfluid helium) and very fast temperature jumps from 1.2 K to ca. 350 K in a few milliseconds. They feature an open porosity up to 80 vol % with pore sizes varying from 150 nm to 2 μm (Fig. 1a), and can contain various concentrations of stable radical species (TEMPO derivatives, 0–285 μmol g$^{-1}$). HYPOPs can be loaded by incipient wetness impregnation (IWI) (Fig. 1b, c) with a variety of liquids ranging from pure pyruvic acid (the gold standard in hyperpolarized metabolic imaging[6]) to complex aqueous solutions. This opens the path towards complex analyses such as mixtures of ligands for fragment-based drug discovery[41], or mixtures of metabolites originating from cell extracts[42]. The $^{1}$H spins of both the HYPOP and the target solution can be polarized in ca. 20 min to very high levels exceeding $P(^{1}H) > 60\%$. This polarization originates from epoxy particles of HYPOP in which PAs are hosted, and spontaneously propagates to the frozen solution by $^{1}$H nuclear spin diffusion (Fig. 1d)[43]. This high $^{1}$H polarization is sheltered on low-gamma $^{13}$C nuclear spins of target molecules after brief CP contacts of a few milliseconds[10]. High levels of $^{13}$C polarization exceeding $P(^{13}C) > 30\%$ can thus be achieved in tens of minutes (Fig. 1d), and further stored at low temperatures (3.8 K) for hours in view of transport to a remote point of use (Fig. 1e). We finally show how the hyperpolarized solutions can be extracted from HYPOPs with high efficiency leading to hyperpolarized solution of metabolites that show very good signal enhancements upon immediate analysis with $^{13}$C NMR (Fig. 1f).

**HYPOP synthesis and characterization.** A considerable number of strategies can lead to polymer networks with open porosity of various sizes[44,45], such as selective degradation of block copolymers assembled in co-continuous morphologies[46], high internal

phase emulsions (HIPE)[47], colloid templating[48], aerogels or open-cell foams[49]. These methods also share some of the following shortcomings: (i) synthetic complexity or incompatibility of polymerization strategies with the presence of stable radicals; (ii) presence of additives in the materials such as surfactants; or (iii) high temperature or chemical treatments incompatible with the survival of the TEMPO radicals. Other strategies, often applied to the synthesis of polymer membranes involve spinodal decomposition between a polymer and solvents through thermal transitions or changes in the solvent composition[50]. The porous network can then be obtained through straightforward removal of the solvents, but often displays significant heterogeneities. We rather opted for an analogous method, where spinodal decomposition between the polymer and the solvent is caused by the polymerization process itself. This process is also relatively easy to implement using thermosetting polymers with superior thermomechanical properties such as epoxy resins. Formulations of epoxy resin, hardener and a non-reactive solvent that display initially full miscibility, and that undergoes phase separation during the curing have been previously studied to form either epoxy materials as dispersed particles[51], solids with closed porosity[52,53], and also solids with open porosities[54].

Thanks to an extensive study of the formulation (nature of the non-reactive solvent and relative fraction to the epoxy resin precursors), we were able to construct a pseudo-phase diagram of such a system and to target network morphologies in close adequacy with our requirements. We selected conventional epoxy resin precursors leading to high $T_g$ thermosets above 150 °C: diglycidyl ether of bisphenol A (DGEBA) and isophorone diamine (IPDA) (Fig. 2a). The mixture was cured at 373 K for 24 h in the presence of polypropylene glycols (PPGs) of different molar mass acting as non-reactive solvents (see detailed synthesis in the "Methods" section). Thorough variations of the fraction of non-reactive solvents (from 30 to 90 wt%) and of their molar mass (from 192 to 2000 g mol$^{-1}$) have been tested in the absence

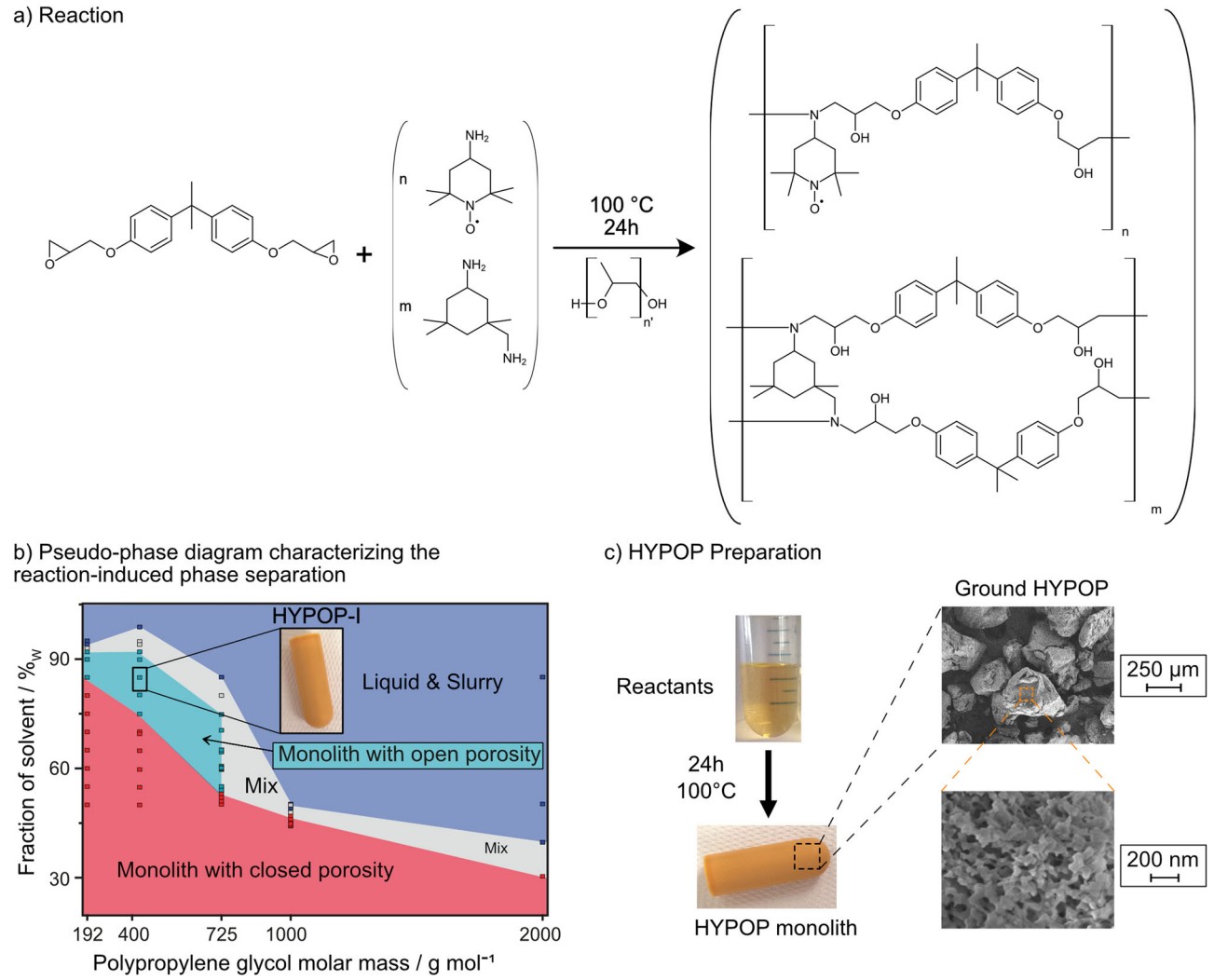

**Fig. 2 Synthesis of HYPOP materials. a** Synthesis of TEMPO-functional and structured epoxy resins from diglycidyl ether of bisphenol A (DGEBA), isophorone diamine (IPDA) and 4-amino-TEMPO in the presence of various amounts of non-reactive polypropylene glycol (PPG). **b** The various morphologies obtained in the absence of 4-amino-TEMPO, when varying the fraction and the average molar mass in number ($\bar{M}n$) of PPGs are reported in a pseudo-phase diagram: liquids (slurry and suspensions) in dark blue, solids with closed porosity in red, heterogeneous mixtures of solids and liquids in grey and solids with open porosity in light blue. The highlighted data (85 wt% of PPG-400 g mol$^{-1}$) corresponds to the HYPOP-**I** series (**c**) that was functionalized with 4-amino-TEMPO and is reported in the remainder of the manuscript.

of 4-amino-TEMPO ($n = 0$). This results in the formation of epoxy particles aggregated in a variety of morphologies such as stable latexes, unstable suspensions, gels, and monolithic networks with closed and opened porosities. The corresponding pseudo-phase diagram, presented in Fig. 2b, was built from a series of experiments (each dot) and the nature of porosity, opened or closed, was assessed by gravimetry after extraction of the PPG (see SI Section 2.2 for details). The functionalization of this resin with commercially available 4-amino-TEMPO is straightforward and was obtained by merely mixing this commercial TEMPO derivative with IPDA in various molar ratios; the amount of DGEBA was adjusted to maintain a proper stoichiometry between amines and epoxides ($n_{\text{epoxy}}/n_{\text{NH}_2} = 2$). Solid samples were characterized, after proper removal of PPG using solvent washes and freeze-drying, by SEM, N$_2$ physisorption and Hg intrusion porosimetry when appropriate. All experimental details are given in the "Methods" section and in Section 2 in the SI.

Spinodal decomposition leading to bicontinuous and homogeneous morphologies, and thus to solids with open porosity after

extraction of the PPG, occurred only in a narrow range of solvent fractions (60–90 wt%), and for low molar mass of PPG. The corresponding size of aggregated epoxy particles was critically dependent on the molar mass of PPG, and varied from 1 to 5 μm, 100 nm and ca. 10 nm for PPG-725, PPG-400 and PPG-192, respectively (see Section 2.4 in the SI). The latter networks were obtained as transparent gels (i.e. not demonstrating extended phase separation between the epoxy and the solvent) and display structures typical of aerogels. Thus, the morphologies of the sample obtained with 85 wt% of PPG-400, showing a bicontinuous morphology composed of ca. 100 nm large aggregated epoxy particles forming a solid with hierarchical porosity ($S_{\text{BET}} = 96$ m$^2$ g$^{-1}$), was particularly well suited to our requirements, with a $T_g$ estimated to be 134 °C and the absence of fracture when immersed in liquid N$_2$. The proper curing of this particular network was monitored with in situ rheology (Section 2.6 in the SI) that enabled the identification of a first phase-separation event after 3 h of reaction, followed 30 min later by the formation of a network between aggregated particles. As the storage modulus reached a plateau after about 15 h of reaction, we kept a curing

time of 24 h for all samples. After subsequent extraction of the PPG and drying of the porous polymers, we found that the final weights of the solids are, within experimental error, the same as the epoxy-amine precursors. In addition, swelling tests indicate the complete absence of a soluble fraction and therefore a complete cross-linking process.

The epoxy formulation using 85 wt% of PPG-400 was modified with increasing amounts of 4-amino-TEMPO (with $r = n_{TEMPO-NH2}/n_{IPDA}$ ranging from 0 to 1.06) to obtain a series of seven HYPOP-I samples that are further used for the dDNP experiments. The concentration of radicals present in the final materials cannot be directly estimated from the initial amount of 4-aminoTEMPO as significant deactivation by disproportionation reaction is occuring[55]. Thus, we quantified the effective concentrations of radicals by Electron Paramagnetic Resonance EPR (see Section 3 in the SI), and found a fairly constant and reproducible survival yield of 34% in comparison to the concentration of 4-aminoTEMPO initially added (Table S2 in the SI). This value, seemingly low, was to be expected from the long curing times (24 h) at 100 °C. This low rate is, however, acceptable in the HYPOP-I series given (a) the relatively low cost of precursors and the simplicity of the synthesis and (b) the large range of radical concentrations that are still attainable, up to 285 µmol g$^{-1}$, which goes beyond optimal concentrations required for dDNP (see below). Incorporating such large amounts of 4-aminoTEMPO in the epoxy networks in place of the tetra-functional IPDA, and therefore at the surface of the pores as well as in the bulk of the material, can yet induce significant changes in the final materials morphology due to (i) the decrease of both cross-link density and $T_g$ in the epoxy network and (ii) changes of solubility parameters between epoxy and PPG that govern the spinodal decomposition and the final morphology of the materials. While an inspection of the HYPOP-I series with SEM (Section 2.5 in the SI) hardly displays evidence of variations in the morphology, a more thorough analysis involving mercury intrusion porosimetry indicates noteworthy changes in the pore size distribution (Section 2.7 in the SI). While all samples display similar distribution of pore sizes in the 10–100 nm range, i.e. within the interstices of the epoxy particles, micrometric pores corresponding to voids between particle aggregates are progressively disappearing when the amount of 4-amino-TEMPO is increased. Concomitantly, the total porous volume decreases from about 3.5 to 1 mL g$^{-1}$ and the surface area determined by nitrogen physisorption from an $S_{BET}$ of 96 to 42 m$^2$ g$^{-1}$ (Fig. S4 in the SI). We believe this to be due to an increased compatibility of the TEMPO-rich epoxy with the PPG, and thus a more extensive plasticization and susceptibility to pore collapse upon removal of the solvent.

**HYPOP impregnation with aqueous solutions**. While HYPOP-I samples could be completely backfilled by immersion in a few centimetres of pure water or aqueous solutions, their low surface energy does not allow for spontaneous capillary impregnation. Due to the high costs of concentrated $^{13}$C-labelled solutions, we rather resorted to use water/ethanol mixtures (9/1 v/v) to enable direct capillary impregnation. Swelling of the HYPOP matrix is another important parameter to consider that could affect the PA concentration either by scavenging the radical[56], or simply by increasing the volume of the polymer and therefore decreasing the radical concentration and the dynamics of polarization. While a few organic solvents demonstrated extensive swelling, the water: ethanol mixtures used in this paper induced only a moderate swelling of 19% (see Table S3 in the SI).

In a final step, the HYPOP-I monoliths were ground and sieved into ca. 250–500 µm powders (Fig. 2c). The initial polarizability of $^1$H spins in the HYPOP-I series was first characterized in the absence of an impregnated solution to optimize the PA concentration necessary for optimum DNP conditions. In the remaining study, wet dDNP samples were obtained by slightly

### a) Polarization ($^1$H) as function of the radical concentration

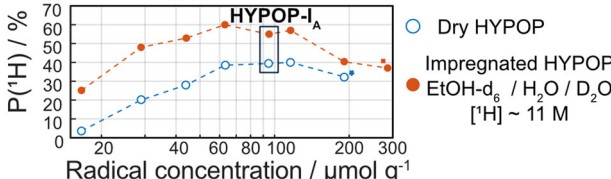

### b) R$_{DNP}$ ($^1$H) as function of the radical concentration

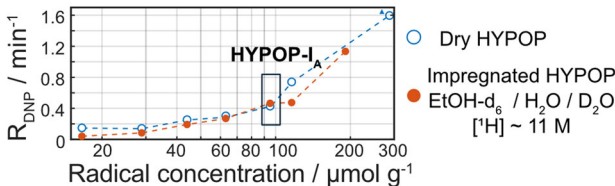

**Fig. 3 $^1$H DNP performances of the HYPOP-I series.** Samples were compared before (blue open circles) and after (red filled circles) impregnation with a solution of ethanol-d$_6$/H$_2$O/D$_2$O (1/1/8 v/v/v), polarized at 1.4 K and 7.05 T, with a microwave frequency of $f_{uw}$ = 197.648 GHz using a triangular frequency modulation of width $\Delta f_{uw}$ = 160 MHz and a modulation rate $f_{mod}$ = 500 Hz. **a** Steady-state $^1$H polarization levels (in case of incomplete build-up, asymptotic polarization values have been extrapolated from the fits in Sections 8.2 and 8.3 in the SI) and **b** corresponding polarization build-up rate $R_{DNP}(^1H) = 1/T_{DNP}(^1H)$ values. *, ■,Δ Please see details in Sections 8.2 and 8.3 in the SI.

under-impregnating 20 mg of HYPOP-I powder with 60 µL of solutions.

**$^1$H polarization with HYPOPs**. DNP was performed at 7.05 T and 1.4 K in a Bruker prototype dDNP polarizer according to a standard DNP protocol described in the "Methods" section. The $^1$H polarization kinetics followed a conventional first-order mono-exponential increase for all samples. Corresponding final $^1$H polarization values and build-up rates are presented in blue in Fig. 3a, b, respectively. A maximal $^1$H polarization of about 40% was reached for HYPOP-I samples containing radicals at concentrations between 60 and 120 µmol g$^{-1}$ while the build-up rates increased continuously with the radical concentration. Such behaviour was previously observed in HYPSO materials, which demonstrated optimal polarizations at radical concentrations between 50 and 100 µmol g$^{-1}$ [36].

After impregnation of HYPOP-I samples with $^1$H concentrations ca. 11 mol L$^{-1}$ (D$_2$O/H$_2$O/ethanol-d$_6$ mixture (8/1/1 v/v/v)), we performed DNP experiments with the same DNP protocol, which led to a similar optimal PA concentration, but significantly higher maximal levels of polarization of about $P(^1H)$ = 55% (Fig. 3a, b, in red dots). This increase in polarization to levels beyond those of dry HYPOP-I samples was unexpected. One possible explanation for this phenomenon would be that the PAs are to some extent heterogeneously distributed across the aggregated particles forming HYPOP-I, some particles exhibiting a lower $^1$H polarization than others and, being connected to the others through a tortuous path, are unable to polarize by long-range $^1$H spin diffusion. However, once the porous polymer is impregnated with a $^1$H containing solution, $^1$H spin diffusion across the whole sample is strongly facilitated, which may help to reach proper hyperpolarization in all parts of the sample. In addition, the $^1$H polarization kinetics diverge significantly from first-order, which indicates heterogeneous build-up kinetics. This feature is typical for DNP build-up curves when polarization in the vicinity of the radicals is coupled to significant long-range spin diffusion into radical-poor domains (the frozen solutions in

our case)[28]. In order to provide comparable build-up rates, the $^1$H polarization kinetics were fitted with stretched exponentials

$$P(^1H) = P_{max}\left(1 - e^{-(R_{DNP}t)^\beta}\right),$$ (1)

with $\beta$ being the breadth of the distribution of build-up rates. The average build-up rate $R_{DNP}^*$ is defined as:

$$R_{DNP}^* = R_{DNP}\frac{\beta}{\Gamma(\frac{1}{\beta})},$$ (2)

with $\Gamma$ the gamma function[57]. Both dry and impregnated matrix build-up rates follow a similar trend, albeit understandably slower for impregnated HYPOP-**I** than for empty materials. The optimal formulation chosen for the rest of this work, HYPOP-**I$_A$** containing radicals at 95 µmol g$^{-1}$, provides fast and extensive $^1$H polarizability to the frozen solution. Such nitroxide concentration is close to the optimal concentration of 50 µmol g$^{-1}$ generally reported under dDNP conditions. Increasing further the concentration of radicals has been reported to accentuate electron spin dipolar couplings and to eventually lead to an electron spin resonance broadening and thus a decrease in final nuclear spin polarization. Polarization values higher than $P(^1H) = 55\%$ might potentially be obtained by partially deuterating the HYPOP material.

**Generating and storing $^{13}$C hyperpolarization in HYPOP-I$_A$.** Firstly, 60.2 mg of HYPOP-**I$_A$** sample was impregnated with 120 µL a D$_2$O:ethanol (9:1 v/v) solution also including 1 M [$^{13}$C]urea, 1 M [1-$^{13}$C]glycine, 1 M [$^{13}$C]sodium carbonate and 1 M [$^{13}$C] sodium formate (four distinct target molecules) and 150 mM of sodium ascorbate. The latter is a reducing agent that readily scavenges paramagnetic oxygen in the aqueous solution as well as the PAs at the surface of the porous polymers and thus attenuates the corresponding paramagnetic relaxations[56]. The quick and efficient $^1$H polarization generated in HYPOP-**I$_A$** and in the impregnated solution, reaching about $P(^1H) = 53\%$ after a 2 min build-up time (Fig. 4a), is efficiently transferred to $^{13}$C spins by $^1$H → $^{13}$C multiple-contact CP repeated every 4 min[10], and leads to a polarization of the $^{13}$C target molecules of about $P(^{13}C)$ ~25% with a 7.8 min build-up time (Fig. 4b). Sequence is detailed in the "Methods" section as well as in Section 4.2 in the SI.

Such performances in the presence of sodium ascorbate in the impregnation solution confirm that a significant part of the radicals is located within the bulk of the HYPOP matrix. Yet, the major innovation in our system does not only consists in high $^{13}$C polarizability, but also and primarily in the extended hyperpolarization lifetime in solid state, $T_1(^{13}C)$. The latter was assessed from the $^{13}$C relaxation at 3.8 K and 7.05 T (Fig. 4c), monitored over 15 h by $^{13}$C detection with small flip angle pulses every 30 min. The pulse angle was chosen to be ~5° prevent excessive perturbation of the hyperpolarization. The relaxation time constant $T_1$ was estimated about 5.7 ± 0.1 h, taking into account the slight depletion of polarization due to the radiofrequency pulses (see Section 8.4 in the SI). This relaxation process is essentially due to paramagnetic relaxations towards the PAs in the bulk HYPOP-**I** through $^{13}$C–$^{13}$C nuclear spin diffusion. In comparison to previous polarization sources for dDNP such as surface-functionalized mesoporous silica[35], or thermoresponsive hydrogels[39], we attribute the dramatically slower relaxations to the low $^{13}$C abundance in the bulk HYPOP (natural abundance), and mostly to the structured morphology of the HYPOP materials featuring pore sizes of around 100 nm and a glassy, relatively hydrophobic matrix that ensures a limited diffusion of $^{13}$C analytes within the bulk HYPOP during the impregnation step. Quantification of the resulting spatial separation between

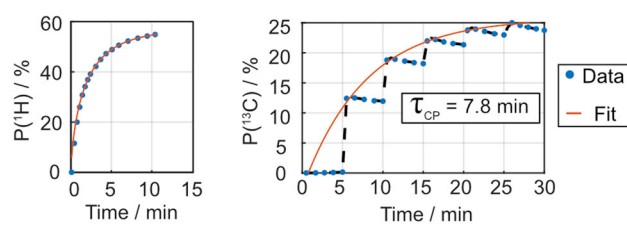

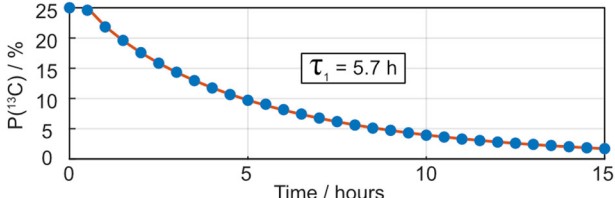

**Fig. 4 $^{13}$C hyperpolarization and solid-state relaxation of target solution in HYPOP-I$_A$. a** $^1$H DNP build-up of HYPOP-I$_A$ sample impregnated with a solution (see in the text) measured at 1.4 K and 7.05 T, with a microwave frequency of $f_{uw}$ = 197.648 GHz with a triangular frequency modulation of width $\Delta f_{uw}$ = 160 MHz and rate $f_{mod}$ = 500 Hz. **b** Subsequent multi-CP transfer of polarization to $^{13}$C of target molecules. **c** Subsequent $^{13}$C relaxation after warming the sample to 3.8 K, showing a characteristic decay constant $T_1(^{13}C)$ = 5.7 ± 0.1 h.

the PAs embedded in the bulk of HYPOP and the $^{13}$C nuclear spins of metabolites in the impregnating solution will be addressed in a forthcoming publication.

Such a long solid-state $T_1$ relaxation time in the case of a solution of target molecules paves the way towards transport of hyperpolarized frozen solutions. As we previously demonstrated, transport can in principle be achieved using a helium Dewar, equipped with an assembly of permanent magnets providing a moderate magnetic field (1 T)[23,28]. Different strategies for fast transfers[23] have been reported, which could also be implemented in the future in combination with our approach.

**Dissolution and analysis of hyperpolarized solutions from HYPOP-I$_A$.** Dissolution was performed by standard dDNP methods[1], using superheated D$_2$O (180 °C) injected directly after CP in the sample holder located in the polarizer as previously described[10]. The impregnated HYPOP powder was flushed out of the cryostat during the process together with the hyperpolarized solution. Therefore, an additional inline filter (see Section 6 in the SI) was used to retain the HYPOP powder while letting through the melted hyperpolarized solution. The extracted hyperpolarized solution was transferred into a benchtop 80 MHz NMR apparatus (Bruker F80) equipped with a dedicated injector (Bruker BioSpin prototype), coupled to a conventional 5 mm NMR tube and ready for analysis. The whole transfer (dissolution, filtration and injection in the NMR tube) up to the start of the NMR pulse sequence took approximately 1.7 s. A magnetic tunnel[23] (4 mT solenoid) was implemented along the transfer capillary to prevent excessive polarization losses at low field, except at the position of the HYPOP filtration device (we believe that minor technological improvements solving this shortcoming may considerably increase the overall performance of the method).

After hyperpolarization of a solution of 1 M [1-$^{13}$C]sodium acetate, 1 M [$^{13}$C]sodium formate and 1 M [$^{13}$C]glycine in HYPOP-**I$_A$**, the immediate dissolution and recording of $^{13}$C spectra gave an intense signal (Fig. 5a, 1 scan) measured on a Bruker Fourier 80 benchtop NMR spectrometer (80 MHz $^1$H

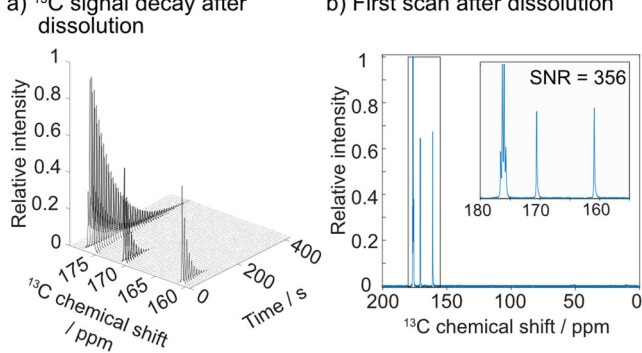

**Fig. 5 NMR spectra of extracted hyperpolarized metabolites. a** Full and detailed view of the first $^{13}$C spectrum. **b** Time series of the $^{13}$C hyperpolarized spectra measured every 5 s with a 5° nutation angle pulse in the liquid state after hyperpolarization of a 100 µL solution of 0.9 M [1-$^{13}$C] glycine at, 0.9 M [1-$^{13}$C]sodium formate, 0.9 M [1-$^{13}$C]sodium acetate, in 90% D$_2$O and 10% ethanol-$d_6$ in HYPOP-I and dissolution with 7 mL of D$_2$O, filtration transfer and injection into a Bruker Fourier 80 benchtop NMR spectrometer (80 MHz $^1$H frequency).

frequency). The asymmetry of the $^{13}$C signals indicates a substantial proton polarization of the J-coupled $^1$Hs[58]. Enhancement factors were calculated (see Section 8.5 in the SI) to be ca. 12'000 for [1-$^{13}$C]-acetate and 6'200 for $^{13}$C formate (i.e. about 2% and 1% absolute $^{13}$C polarization, respectively). Time-resolved $^{13}$C spectra were measured every 5 s with 5° (Fig. 5b). $T_1(^{13}$C) relaxation time in liquid of about 90 s for sodium acetate and of about 24 s for sodium formate were observed. These values are within the range expected for $^{13}$C polarization in the absence of paramagnetic electron spins[59], and demonstrate the efficient removal of HYPOP particles upon filtering. Surprisingly, $^{13}$C-glycine signals could not be observed, which indicates a complete loss of polarization upon transfer. The $^{13}$C glycine polarization loss is not related to the usage of HYPOP-I$_A$, but happens in low-field regions during transfer, more precisely in the filtration system that could not be embedded in our magnetic tunnel. Such low fields are indeed known to favour scalar relaxation on $^{14}$N[60]. Moreover, we observed the same behaviour for $^{13}$C-glycine upon dDNP of a standard frozen sample formulations doped with TEMPOL (no HYPOP). The final polarization values reported here do not yet match state-of-the-art values obtained with conventional dDNP sample formulation (ca. 50%[10]), and a lengthy work of optimization is certainly still pending. However, this work illustrates the great potential of dDNP using HYPOPs as polarization generation, storage and transport matrices, which paves the way to a widespread use of dDNP in NMR or MRI experiments. Further technological improvements on the transfer line (e.g. magnetic tunnel over the whole line) still need to be implemented and may help to further increase the performance of the method.

In this paper, we have demonstrated that porous polymers (HYPOP-I) containing TEMPO derivatives as PAs and featuring a porosity spanning from 50 nm to 1 µm can be used as multipurpose matrices for dDNP applications, with the antagonistic properties of rapidly generating $^{13}$C hyperpolarization in frozen aqueous mixtures, while preserving this high polarization for extended lifetimes of several hours in view of transport experiments. Together with the facile extraction of pure hyperpolarized solutions of target molecules, our strategy paves the route to a broader democratization of the use of dDNP in NMR or MRI experiments. Our strategy was illustrated with porous epoxy resins that combine straightforward synthesis and tunable morphologies, and we anticipate that this concept of

PA/target molecules' separation will be further generalized to a variety of porous solids, with potentially further improved performances.

## Methods

**HYPOP synthesis and preparation.** In a polypropylene tube, diglycidyl ether of bisphenol A (DGEBA), isophorone diamine (IPDA), amino TEMPO were mixed with polypropylene glycol (PPG) as a solvent. Amount of DGEBA was calculated in order to maintain a ratio 2/1 with primary amines of IPDA and amino TEMPO. Quantity of amino TEMPO was calculated to reach a desired concentration taking into account the survival yield of the synthesis and the quantity of solvent was fixed at 85% of the total weight of the solution. Solutions were mixed and degassed under partial vacuum (>0.01 mbar). Heating the sample during this operation with a heat gun (<100 °C) can accelerate the process. Polymerization took place at 100 °C for 24 h. PPG was finally extracted by 3 exchanges with a large quantity of ethanol followed by 3 exchanges with water during at least 3 h each before freeze-drying them. Further details of the synthesis, including effects of PPG size on porosity, are given in Section 2 in the SI. HYPOP-I monoliths are manually ground and sifted to select particles between 250 and 500 µm.

**Chemicals supply.** All chemicals used (reactants, monomers, solvents, analytes, etc.) are currently commercially available. The list of suppliers is accessible in Section 1 in the SI.

**Scanning electron microscopy.** Scanning electron microscopy experiments were performed on a ZEISS Merlin Compact or on a Quanta FEI 250 after deposition of 10 nm of copper using secondary electrons detectors.

**Polarizer apparatus.** All DNP measurements were performed with a prototype dDNP Polarizer developed by Bruker BioSpin, with a helium bath cryostat operating within a range of 1.2 K < T < 4.2 K at a magnetic field of 7.05 T and equipped with $^1$H/$^{13}$C homemade NMR probe. Microwaves were generated with a Virginia Diode system (8–20 GHz VDI synthesizer with a 198 GHz AMC amplifier/multiplier chain). For the characterization of HYPOP, a KelF sample holder was used to reduce the $^1$H signal background. For dissolution experiments, a more robust PEEK sample holder was used.

**$^1$H DNP experiments.** Prior to DNP experiments, HYPOP powders were used as is or impregnated with solutions to be hyperpolarized, placed in the sample cup, and inserted in the liquid helium bath of the dDNP polarizer at 4.2 K. $^1$H NMR signals were measured at 3.8 K (700 mbar pressure), with 0.1° pulses every 5 s during 10 min until reaching the thermal equilibrium plateau. The quality factor of the $^1$H NMR circuit was by attenuated by adding a 50 Ohm resistor in series with the tuning and matching box, so as to decrease radiation-damping (potentially very intense at high polarization values), which can lead to underestimation of the polarized $^1$H signal. $^1$H polarization curves were recorded at 1.4 K and 7.05 T, with the following microwave parameters: frequency of $f_{uw} = 197.648$ GHz, triangular frequency modulation of width $\Delta f_{uw} = 160$ MHz, modulation rate $f_{mod} = 500$ Hz, and estimated power in the sample cavity $P_{uw} = 30$ mW.

**$^{13}$C DNP experiments.** After inserting the sample, $^{13}$C NMR signals were measured with 5° pulse every 30 min during 6 h at 3.8 K and once the plateau was reached, the thermal equilibrium NMR signal was recorded with a series of 64 pulses for improved sensitivity (see Section 8.4 in the SI). The same procedure was previously applied for the measurement of the $^{13}$C background signal (without sample). $^1$H → $^{13}$C CP was performed using a 6 ms contact every 5 min to allow time for $^1$H polarization to build-up and diffuse in the frozen solution in the pores of HYPOP. The CP matching condition was realized with 23 kHz $B_1$ field on both $^1$H and $^{13}$C channels (with 8 ms square pulse of $^1$H (8 W) and an amplitude of $^{13}$C pulse ramped linearly from 50% to 100% (150 W at maximum). Adiabatic half-passage pulses (100 kHz, 175 µs with 12 W for the $^1$H channel and 150 W for the $^{13}$C channel) were used to flip the $^1$H and $^{13}$C magnetization in the transverse plane before CP contact and to restore magnetization along z afterwards.

**Dissolution, transfer and injection experiments.** A solenoid of 1.5 m length has been built around the transfer capillary, and fed with a current of 2 A, thus generating a 4 mT field all along the transfer line except for the filtering system placed in close proximity to the benchtop spectrometer. First, 7 mL of D$_2$O solvent was pressurized at 6 bar and heated to reach 9 bar. After the dissolution, the HYPOP-I matrices mixed together with the molecules of interest are pushed with hot D$_2$O through a filtering device described in Section 6 in the SI. Finally, the solution was fed into a Bruker's prototype NMR injector directly placed in the Fourier80 benchtop NMR spectrometer.

**Hyperpolarized liquid-state NMR measurements.** Liquid-state hyperpolarized NMR spectra were measured in a Fourier 80 MHz Bruker BioSpin benchtop

spectrometer ($^1$H frequency 80.222 MHz and $^{13}$C frequency 20.1718 MHz). After injection, $^{13}$C signals were recorded every 5 s with a 5° nutation angle pulse. Final hyperpolarization enhancement was calculated by cross-calibration with a highly concentrated 3 M [1-$^{13}$C]sodium acetate reference sample, with a simple method described in detail in Section 8.5 in the SI.

## Data availability

The characterization of porosity, RPE and raw NMR data used in this study is freely available in the Materials Cloud archive: 10.24435/materialscloud:kv-6q.

## Materials availability

The materials of this study are available from the corresponding authors upon reasonable request.

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

## Acknowledgements
In memory of Prof. Jean-Pierre Pascault. This research was supported by ENS-Lyon, the French CNRS, Lyon 1 University, the Institute of Chemistry at Lyon (ICL), the European Research Council under the European Union's Horizon 2020 research and innovation program (ERC Grant Agreements No. 714519 / HP4all and Marie Skłodowska-Curie Grant Agreement No. 766402 / ZULF). The authors acknowledge Bruker BioSpin for providing the dDNP system and for their support; Venita Decker and Frank Decker from Bruker BioSpin for providing the F80 spectrometer and for their support; David Gajan, Arianna Ferrari, and Stuart J. Elliott for their assistance on the dDNP apparatus; Xiao Ji, Fanny Russel, Adrien Alonzo and Alexandra Erdmann for their synthesis work; Catherine Jose and Christophe Pages from the "service prototype" of the "Institut des Sciences Analytiques"; Stephane Martinez from the "Atelier de mecanique de l'UCBL". Emilien Etienne and Bruno Guigliarelli from the interdisciplinary French EPR network Renard at Univ. Aix. Marseille. The "Centre Technologique des Microstructures" (CTμ) and Pierre-Yves Dugas for their help with SEM.

## Author contributions
T.E.D. and D.M. designed and synthesized HYPOP materials. T.E.D., L.G. and D.M. characterized HYPOP materials. S.F.C., T.E.D., O.C., Q.S., A.B., J.M. and B.V. performed the dDNP experiments. T.E.D., S.F.C. and Q.S. analysed the DNP results. M.C., J.K., R.M., D.E. and M.S. designed and built the dDNP hardware. S.J. designed the project and supervised the team effort. S.F.C., T.E.D., S.J. and D.M. wrote the manuscript.

## Competing interests
The authors declare no competing interests.
