## [Peer Review File. · Nature Communications]

Reviewers' comments:

Reviewer #1 (Remarks to the Author):

This manuscript reports new porous polymeric materials that are functionalized with stable paramagnetic species that enable them to be effectively used to hyperpolarize ^{13}C nuclei to high extents (ca. 25%) and which can furthermore be preserved for up to several hours at liquid helium temperatures. The significance of this is that it extends the time over which extremely high levels of ^{13}C nuclear spin polarization (0.25 compared to ca. 10^{-4} at room temperature) can be stored for later use in spectroscopic or imaging applications. The manuscript is well-written and the results are interesting and of potentially high impact. The Introduction is pedagogical, which may be helpful for a broad audience, though is longer than it needs to be and could be condensed. Overall, the manuscript reports new and interesting advancements in materials chemistry and in physical characterization methodology that make it likely suitable for publication, after the following points are addressed:

1. The Abstract contains mostly non-scientific commentary that focuses on motivational aspects of the work, with little discussion of the main results. For a scientific journal like Nature Communications, the abstract should emphasize principally the new scientific results that are presented in the manuscript.

2. Page 2, column 2, lines 12-13: The authors state "thanks to thorough investigation," the source of which is unclear. Being in the Introduction, the phrase suggests that such investigation was previously reported in the literature, in which case references should be cited. Or if it pertains to the current manuscript, then it sounds self-congratulatory. This should be clarified.

3. Page 3, Figure 1: Figure 1 has some degree of pedagogical value but its contents are widely known in the field and not new. In the caption: (a) the HYPOP material has not yet been described. What is its composition and structure? More description is needed. (b) A length scale bar is needed in Fig. 1b.

4. Page 4, column 1, lines 8-9 from bottom: The long (5.7 h) ^{13}C spin-lattice relaxation time is impressive, but other than the low temperature (3.8 K), little information is provided. For what specific molecules, solvent (partially deuterated?), and HYPOP-I compositions? Such information is necessary for others to be able to reproduce or appreciate the work and the manuscript should not be published without such information and discussion.

5. Page 5, column 2, middle: How was a "bicontinuous morphology" established? What is the surface area of the HYPOP matrix?

6. Page 5, column 2, line 5: Figure 2b is helpful and interesting. However, it should not be called a "phase diagram," as it does not reflect systems at equilibrium, due to the crosslinking reaction of the epoxy resin.

7. Page 5, column 2, lines 1-2 from bottom: The quantitative estimate that 34% of the radicals survive the synthesis conditions is good to have assessed. Where are the TEMPO groups? At the surface, embedded in the bulk HYPOP matrix, and to what extents? What accounts for the assembly of the TEMPO molecules at the surface or becoming embedded within the porous polymer? Such understanding is important to provide, as such considerations appear to be central to the manuscript's main conclusions.

8. Page 6, column 2, Figure 3: Figure 3a is interesting. What a RDNP needs to be defined and Figure 3b is not discussed.

9. Page 7, column 1, line 5: The authors should explain what factors account for the optimum value of 95 $\mu\text{mol/g}$ radicals in HYPOP-I. What considerations might be undertaken to improve performance further?

10. Page 7, Figure 4 and column 1, lines 7-8 from bottom: Figure 4 is interesting and informative

and the very long ^{13}C T1 is extraordinary. What are the properties of the HYPOP material(s) that explain this very long value?

11. Page 8, column 1, lines 1-5: The authors state that the ^{13}C signals from glycine were not observed, and they speculate that these moieties were completely relaxed during low-field transfer. Why would they be selectively relaxed in the filtration system or by any other means, compared to other ^{13}C species?

12. Page 8, Conclusions: After the emphasis in the Introduction on transportability and opportunities for analyses in locations distant from the polarizing laboratory, the manuscript stops short of conducting meaningful measurements that would demonstrate the feasibility of these practical aspects. I recommend that the Abstract and Introduction be written to focus more on the HYPOP material and understanding how its composition and structure leads to its favorable hyperpolarization properties and long ^{13}C spin-lattice relaxation times.

13. Page 9, middle: It is noteworthy that the high ^{13}C signal sensitivity and resolution was conducted under low-field (1.9 Tesla) conditions.

Reviewer #2 (Remarks to the Author):

The article by El Daraï et al. entitled "Porous Functionalized Polymers enable Generating and Transporting Hyperpolarized Arbitrary Solutions" is a very interesting paper, both on a conceptual and on a scientific results point-of-view, that I have read with great curiosity and pleasure.

The issue addressed by the authors, i.e. the democratization of the DNP technique, is a key modern problematic: Shall we wait that the scientific community raise funds to buy DNP equipment's or shall we think to provide keys to bring DNP benefits into laboratories? The proof-of-concept brings by the authors is very promising. This article is well worth publishing in Nat. Comm. Here some comments and questions in order to clarify some aspects of the paper.

1) Several times, the expression "arbitrary solutions" comes up again and again in the manuscript. However, the only given example is a polar and protic solution of H_2O /ethanol mixture.

2) To further increase the interest of the approach, it would be extremely interesting to know the possibility of reusing HYPOP. Have the authors tried some turn-over with HYPOP? Is there a loss of efficiency?

3) One concern that could be crucial to the development of the approach is the fact that it consumes a lot of radicals as only 35% are incorporated in the polymer scaffold. Do the authors have any ideas on how to improve encapsulation efficiency?

4) There are mismatches between reported data in Fig. 3a and data given in Supp Info (8.2 and 8.3 sections). For the dry HYPOP $26 \mu\text{mol.g}^{-1}$ sample, the P(1H) value is found to be 15.21% however it is reported around 20% in Fig. 3a. For the Impregnated HYPOP $45 \mu\text{mol.g}^{-1}$ sample, the value given is 47.32%, however more than 50% is reported in Fig. 3a.

5) I do not really understand why authors ground and sieved their materials in a specific range. Would it be interesting to use the monolith for dissolution manipulation? The filtration part of the process could easier?

6) Still concerning the dissolution part of the work, there is no mention about the delay between the hyperpolarization and the dissolution (i.e. the storage time). Implicitly, the dissolution is done right after hyperpolarization. But, it could be interesting to know if storage (i.e. ^{13}C relaxation at 3.8K) has an influence on the hyperpolarized measured intensity (even if the relationship should be linear).

7) Why the molecules mixture and their concentrations are different between the Hyperpolarization

part and Dissolution part of the manuscript? In the latter part, the concentrations given (0.95, 0.93, 0.92 M) are corrected from the addition of 7 mL superheated D₂O? Typical what is volume extracted from the dissolution procedure? Some liquid is retained by the HYPOP?

8) I do not understand the ¹³C spectrum given in Fig. 5a. The two signals on the left might correspond to acetate and formate respectively, but what about the peak around 160 ppm? Could the author provide an assignment?

9) Surprisingly, Glycine is absent on the ¹³C spectrum. Could we imagine a specific adsorption of glycine on the polymer surface? More generally, do the authors consider that some category of molecules could be retained by the porous polymeric matrix?

10) The final radical concentration values given in Table S1 (17, 29, 44, 63, 95, 116, 191, 286 μmol.g⁻¹) are not consistent with the rest of the manuscript (16, 26, 45, 62, 93, 116, 192, 285 μmol.g⁻¹).

11) Overall, the quality of the SEM images is really poor, especially those at high magnification.

12) In Fig S10, we do not really understand that is a "10 M 1H solution". Moreover, the impregnated and Dry samples, that are compared, are not of the same amount of radical.

Dr. Thierry Azaïs

Reviewer #3 (Remarks to the Author):

This paper presents a new tempo-functionalized epoxy based polymer formulation as a polarising agent for dissolution DNP experiments. The formulations are shown to perform well for the polarization of model solutions (but at this stage not better than previous work, as the authors acknowledge), and the paper provides a detailed description of the optimisation of the polymer and the sample formulation.

The work well carried out and is a nice addition to the library of formulations for dissolution DNP, and the formulations introduced here have some features that are advantageous. However, the work builds on and combines the authors' own extensive previous work using either porous silica networks or microcrystalline formulations.

It is surprising that there is little to no mention of previous work using polymer networks as polarisation sources, which have been envisaged in a range of contexts for at least the last 10 years. An especially relevant early example is Phys. Chem. Chem. Phys., 2010,12, 5879-5882. Others include Macromolecules 2018, 51, 20, 8046–8053; Journal of Magnetic Resonance 190, 2008, 307-315; J. Am. Chem. Soc. 2018, 140, 22, 6969–6977; A quick search even yields an epoxy-tempo mixture <https://doi.org/10.1016/j.nima.2014.11.114>

The article "Producing Radical-Free Hyperpolarized Perfusion Agents for In Vivo Magnetic Resonance Using Spin-Labeled Thermoresponsive Hydrogel" from 2016, Macromol. Rapid Commun. 2016, 37, 1074–1078, which has a similar concept and objective to the present work, is mentioned as reference 33, but it is discounted in three words as "preventing hyperpolarization transport". In reality the work in reference 33 achieves the same separation between the PA and the analyte that is what (conceptually) enables transport here, but by a change in shape of the PA.

With this ensemble of background work in mind, the results here do not represent the kind of conceptual step forward I would expect from an article in Nature Communications.

On a more minor note, the authors refer to polarization of "arbitrary" solutions. My reading is that only water:ethanol was used here, and that pure water was not suitable, and some organic solvents swelled the polymer. Only two quite similar samples are then given which are (i) 1M [¹³C]urea, 1M [1-¹³C]glycine, 1M [¹³C]sodium carbonate, 1M [¹³C]sodium formate and 150 mM of sodium ascorbate; and (ii) 0.9 M [1-

^{13}C glycine, 0.9 M $[1-^{13}\text{C}]$ sodium formate, 0.9 M $[1-^{13}\text{C}]$ sodium acetate. This is anything but "arbitrary". The ascorbate is included as a radical scavenger, and the others are all chosen to have slowly relaxing ^{13}C centers. Of these the glycine, which is the only one with a non-exchangeable CH_2 group, is not observed! There is nothing wrong with this example, but it cannot be used to support the claim that arbitrary solutions can be polarized.

Reviewer #1:

This manuscript reports new porous polymeric materials that are functionalized with stable paramagnetic species that enable them to be effectively used to hyperpolarize ^{13}C nuclei to high extents (ca. 25%) and which can furthermore be preserved for up to several hours at liquid helium temperatures. The significance of this is that it extends the time over which extremely high levels of ^{13}C nuclear spin polarization (0.25 compared to ca. 10^{-4} at room temperature) can be stored for later use in spectroscopic or imaging applications. The manuscript is well-written and the results are interesting and of potentially high impact. The Introduction is pedagogical, which may be helpful for a broad audience, though is longer than it needs to be and could be condensed. Overall, the manuscript reports new and interesting advancements in materials chemistry and in physical characterization methodology that make it likely suitable for publication, after the following points are addressed:

Response:

We thank referee #1 for his/her interest in our work and the positive comments. We provide below answers to each issues that have been raised.

Q1. The Abstract contains mostly non-scientific commentary that focuses on motivational aspects of the work, with little discussion of the main results. For a scientific journal like Nature Communications, the abstract should emphasize principally the new scientific results that are presented in the manuscript.

R1. We have rewritten the abstract, which focuses now essentially on technical performances enabled by the HYPOP materials.

Q2. Page 2, column 2, lines 12-13: The authors state “thanks to thorough investigation,” the source of which is unclear. Being in the Introduction, the phrase suggests that such investigation was previously reported in the literature, in which case references should be cited. Or if it pertains to the current manuscript, then it sounds self-congratulatory. This should be clarified.

R2. We indeed referred to our work in the present manuscript (that includes over 50 experiments to build the diagram in Fig. 2b). We clarified this and made it less “self-congratulatory”.

Q3. Page 3, Figure 1: Figure 1 has some degree of pedagogical value but its contents are widely known in the field and not new. In the caption: (a) the HYPOP material has not yet been described. What is its composition and structure? More description is needed. (b) A length scale bar is needed in Fig. 1b.

R3. We modified the figure according to the suggestion (More scales plus a picture added in part a). We added technical specifications in the figure in order to combine the pedagogical value (which we think is essential for journals with large audience such as Nature Comm.) and the main performances for specialists.

Q4. Page 4, column 1, lines 8-9 from bottom: The long (5.7 h) ^{13}C spin-lattice relaxation time is impressive, but other than the low temperature (3.8 K), little information is provided. For what specific molecules, solvent (partially deuterated?), and HYPOP-I compositions? Such information is necessary for others to be able to reproduce or appreciate the work and the manuscript should not be published without such information and discussion.

R4. The overview was intended to give a short description of the new features distinctive of the HYPOP polymers before giving further details in the remaining of the manuscript. We

combined it within the introduction for more clarity, and thus also shortened the manuscript as advised.

Q5. Page 5, column 2, middle: How was a “bicontinuous morphology” established? What is the surface area of the HYPOP matrix?

R5. The morphologies of porous networks were assessed from gravimetric considerations after extraction of the PPG and sol fraction : we considered that bicontinuous materials are monoliths displaying nearly complete conservation of mass (> 95wt%) between the initial epoxy precursors and the final materials (extracted and dried). The “closed porosity “ region to monoliths that display incomplete extraction of PPG (mass conservation >> 100 wt%). The “mix” region correspond to a highly unrepeatable region where mass conservation is not a useful criterion.

The details of this procedure have been added in section 2.2 in the Supporting Information.

The surface area of HYPOP polymers estimated from the BET analysis of Nitrogen physisorption experiments has been added in section 2.8.

Q6. Page 5, column 2, line 5: Figure 2b is helpful and interesting. However, it should not be called a "phase diagram," as it does not reflect systems at equilibrium, due to the crosslinking reaction of the epoxy resin.

R6. Thank you for this correction. Analogous representations of non-equilibrium self-assembly in polymers are often abusively referred to as “phase diagrams” in the literature. We now refer to Fig 2b as a “pseudo-phase diagram”.

Q7. Page 5, column 2, lines 1-2 from bottom: The quantitative estimate that 34% of the radicals survive the synthesis conditions is good to have assessed. Where are the TEMPO groups? At the surface, embedded in the bulk HYPOP matrix, and to what extents? What accounts for the assembly of the TEMPO molecules at the surface or becoming embedded within the porous polymer? Such understanding is important to provide, as such considerations appear to be central to the manuscript’s main conclusions.

R7. The repartition of TEMPO in the bulk of the porous polymers and not only at the surface is indeed of utmost importance in the performances as DNP polarizing medium, and more particularly in the extended ¹³C spin-lattice relaxation at 3.8K. Although the synthesis route may seem to favour a homogeneous distribution of TEMPO derivatives in the bulk epoxy, one cannot completely rule out specific interactions with the PPG solvent that may drive the TEMPO derivatives at the surface of pores during the phase separation process.

A direct measure of TEMPO concentration as a function of depth from the surface of pores is out of reach even with advanced equipment for surface analysis (e.g. dynamic-SIMS or LIBS). Currently, we are however in the process of measuring indirectly this TEMPO profile by quantifying the DNP performances after loading the HYPOPs with ascorbic acid solutions. The diffusion of ascorbic acid within the bulk epoxy matrix of HYPOP at room temperature gradually scavenges the TEMPO radicals. This ongoing study is however out of scope in the present manuscript and will be the subject of a forthcoming publication.

The most compelling argument that we can presently give in favor of a homogeneous distribution of TEMPO radicals within the bulk of HYPOP matrix is that addition of 150 mM sodium ascorbate in the target solution, which is effectively known to scavenge all radicals located at the surface, does not prevent from performing efficient DNP.

We added a sentence to clarify this: “Such performances in the presence of sodium ascorbate in the impregnation solution confirm that a significant part of the radicals is located within the bulk of the HYPOP matrix”.

Q8. Page 6, column 2, Figure 3: Figure 3a is interesting. What a RDNP needs to be defined and Figure 3b is not discussed.

R8. The definition of R_{DNP} was given in the equation defining the 1H buildup. It is now also given in the caption of Figure 3.

Q9. Page 7, column 1, line 5: The authors should explain what factors account for the optimum value of 95 $\mu\text{mol/g}$ radicals in HYPOP-I. What considerations might be undertaken to improve performance further?

R9. We added the following sentences:

“Such nitroxide concentration is close to the optimal concentration of 50 $\mu\text{mol g}^{-1}$ generally reported in frozen solutions. Increasing further the concentration of radicals has been reported to accentuate electron spin dipolar couplings and to eventually lead to an electron spin resonance broadening and thus a decrease in final nuclear spin polarization. Polarization values higher than $P(1H)=55\%$ might potentially be obtained by partially deuterating the HYPOP material.”

Q10. Page 7, Figure 4 and column 1, lines 7-8 from bottom: Figure 4 is interesting and informative and the very long ^{13}C T1 is extraordinary. What are the properties of the HYPOP material(s) that explain this very long value?

R10. We have rewritten the paragraph as: “This relaxation process is essentially due to paramagnetic relaxations towards the PAs in the bulk HYPOP-I through ^{13}C - ^{13}C nuclear spin diffusion. In comparison to previous polarization sources for dDNP such as surface-functionalized mesoporous silica,³⁴ or thermoresponsive hydrogels,³⁷ we attribute the dramatically slower relaxations to the low ^{13}C abundance in the bulk HYPOP (natural abundance), and mostly to the structured morphology of the HYPOP materials featuring pore sizes around 100 nm and a glassy, relatively hydrophobic matrix that ensures a limited diffusion of ^{13}C analytes within the bulk HYPOP during the impregnation step. Quantification of the resulting spatial separation between the PAs embedded in the bulk of HYPOP and the ^{13}C nuclear spins of metabolites in the impregnating solution will be addressed in a forthcoming publication. »

Q11. Page 8, column 1, lines 1-5: The authors state that the ^{13}C signals from glycine were not observed, and they speculate that these moieties were completely relaxed during low-field transfer. Why would they be selectively relaxed in the filtration system or by any other means, compared to other ^{13}C species?

R11. The polarization losses near zero field depend strongly on the molecular structure and nuclear spin coupling networks. We modified the end of the paragraph to better explain that this is not related to the use of HYPOP:

“Surprisingly, ^{13}C -glycine signals could not be observed, which indicates a complete loss of polarization upon transfer. As we previously observed a similar behavior for ^{13}C -glycine upon dDNP of standard frozen sample formulations, we believe that ^{13}C glycine polarization loss is not related to the usage of HYPOP-IA, but most likely happens in low-field regions during transfer, more precisely in the filtration system that could not be embedded in our magnetic tunnel. Such low fields are indeed known to favour scalar relaxation on ^{14}N .⁵⁹”

Q12. Page 8, Conclusions: After the emphasis in the Introduction on transportability and opportunities for analyses in locations distant from the polarizing laboratory, the manuscript stops short of conducting meaningful measurements that would demonstrate the feasibility of these practical aspects. I recommend that the Abstract and Introduction be written to focus more on the HYPOP material and understanding how its composition and structure leads to its favorable hyperpolarization properties and long ^{13}C spin-lattice relaxation times.

R12. We have rewritten the abstract and reorganized the introduction to include a more thorough description of HYPOP features, comparison of ^{13}C relaxation times with other polarization media previously described in dDNP applications. (“Many materials [...] -> End of introduction)

Q13. Page 9, middle: It is noteworthy that the high ^{13}C signal sensitivity and resolution was conducted under low-field (1.9 Tesla) conditions.

R13. We added a note about it in the main text:

“immediate recording of ^{13}C spectra gave an intense ^{13}C spectrum (Fig 5a, 1 scan) measured on a Bruker Fourier 80 benchtop NMR spectrometer (80 MHz ^1H frequency)”

Reviewer #2:

The article by El Daraï et al. entitled “Porous Functionalized Polymers enable Generating and Transporting Hyperpolarized Arbitrary Solutions” is a very interesting paper, both on a conceptual and on a scientific results point of view, that I have read with great curiosity and pleasure.

The issue addressed by the authors, i.e. the democratization of the DNP technique, is a key modern problematic: Shall we wait that the scientific community raise funds to buy DNP equipment's or shall we think to provide keys to bring DNP benefits into laboratories? The proof-of-concept brings by the authors is very promising. This article is well worth publishing in Nat. Comm. Here some comments and questions in order to clarify some aspects of the paper.

Response:

We thank referee #2 for his interest in our work and the positive comments. We provide below answers to each issues that have been raised.

Q1) Several times, the expression “arbitrary solutions” comes up again and again in the manuscript. However, the only given example is a polar and protic solution of H₂O/ethanol mixture.

R1. The only reason why we used a mixture of H₂O and ethanol (9:1) is to facilitate the impregnation of the HYPOP by slightly decreasing the surface tension of the solution and enabling capillary impregnation. We emphasize that the application of a very slight external pressure (e.g. dipping the HYPOP in a few centimetres of solution) is sufficient to reach complete impregnation in absence of ethanol. We kept using ethanol to avoid handling and wasting large amounts of costly ¹³C-labelled solutions.

We agree however that “arbitrary solutions” is an overstatement, as organic solutions that swell extensively the epoxy matrix should probably not be used (e.g. DCM, DMSO, See the list in Table S3.).

We replaced “arbitrary solutions” by “solutions of metabolites”.

Q2) To further increase the interest of the approach, it would be extremely interesting to know the possibility of reusing HYPOP. Have the authors tried some turn-over with HYPOP? Is there a loss of efficiency?

R2. We agree that reusing HYPOPs is an attractive idea, especially if combined with proper design of monolithic polymers (see below). We do not foresee any loss of PA efficiency upon reusing HYPOPs and are indeed planning to do so in the future.

Q3) One concern that could be crucial to the development of the approach is the fact that it consumes a lot of radicals as only 35% are incorporated in the polymer scaffold. Do the authors have any ideas on how to improve encapsulation efficiency?

R3. The main limitation regarding the final concentration of radicals does not lie with the incorporation of amino-TEMPO during the polymerization, but rather with redox disproportionation reactions (into nitrosonium and hydroxylamine species). We indeed checked i) that the extracted solutions did not contain radical-active species using EPR and ii) that pure solutions of amino-TEMPO in PPG also display significant disproportionation in

comparable curing conditions. We are currently working on limiting the side reactions by controlling the pH during the polymerization.

We have modified the text and added a corresponding reference: “The concentration of radicals present in the final materials cannot be directly estimated from the initial amount of 4-aminoTEMPO as significant deactivation by disproportionation reaction is occurring.⁵⁴ Thus, we quantified the effective concentrations of radicals by Electron Paramagnetic Resonance EPR (See section 3 in SI), and found a fairly constant and reproducible survival yield of 34 % in comparison to the concentration of 4-aminoTEMPO initially added (Table S2 in SI).”

Q4) There are mismatches between reported data in Fig. 3a and data given in Supp Info (8.2 and 8.3 sections). For the dry HYPOP 26 $\mu\text{mol.g}^{-1}$ sample, the P(1H) value is found to be 15.21% however it is reported around 20% in Fig. 3a. For the Impregnated HYPOP 45 $\mu\text{mol.g}^{-1}$ sample, the value given is 47.32%, however more than 50% is reported in Fig. 3a.

R4. We indeed forgot to mention that in order to properly compare maximal polarization values in the case of very different build-up kinetics, we decided to report in Figure 3a the theoretical maximal polarization values, extrapolated from the 1H build-up fits in section 8.2. and 8.3 We added:

“In case of incomplete build up, asymptotic polarization values have been extrapolated from the fits in Section 8.2 and 8.3 in SI”.

In the two example mentioned, it is clear that build up curves did not reach a plateau and final values of polarization would be significantly underestimated by considering only the last data point.

Q5) I do not really understand why authors ground and sieved their materials in a specific range. Would it be interesting to use the monolith for dissolution manipulation? The filtration part of the process could be easier?

R5. Indeed, it was originally our plan to use the monoliths as both polarizing media and filtration devices. Unfortunately, the small porosity in these monoliths induces a dramatic pressure drop which is absolutely not compatible with the currently available pressure for the heated water (9 bars) and the very quick dissolution required in these experiments.

Furthermore, working with calibrated powder helps to increase repeatability of filtration process and avoid frustrating tubes collapsing during the dissolution.

Q6) Still concerning the dissolution part of the work, there is no mention about the delay between the hyperpolarization and the dissolution (i.e. the storage time). Implicitly, the dissolution is done right after hyperpolarization. But, it could be interesting to know if storage (i.e. ^{13}C relaxation at 3.8K) has an influence on the hyperpolarized measured intensity (even if the relationship should be linear).

R6. We have indeed specified that the dissolution was done right after hyperpolarization: “After hyperpolarization of a solution of 1 M [^{13}C] sodium acetate, 1 M [^{13}C] sodium formate and 1 M [^{13}C] glycine in HYPOP-IA, the immediate dissolution and recording of ^{13}C spectra gave an intense ^{13}C spectrum (Fig 5a, 1 scan) measured on a Bruker Fourier 80 benchtop NMR spectrometer (80 MHz ^1H frequency).

By lack of time and equipment availability, we did not carry out the suggested experiment (e.g. carrying out multiple dissolution experiments while varying the delay time between hyperpolarization and dissolution). We agree however that the effective “yield of dissolution”

should remain constant, and that the effective enhancement in liquid state should be directly proportional to the ^{13}C polarization in solid state (Figure 4c)

Q7) Why the molecules mixture and their concentrations are different between the Hyperpolarization part and Dissolution part of the manuscript? In the latter part, the concentrations given (0.95, 0.93, 0.92 M) are corrected from the addition of 7 mL superheated D_2O ? Typical what is volume extracted from the dissolution procedure? Some liquid is retained by the HYPOP?

R7. The quantification of ^{13}C spin lattice relaxation was carried out with a mixture of urea, glycine, sodium carbonate and sodium formate in order to highlight the possibilities of hyperpolarization storage.

Upon repeating the experiment with the same mixture for dissolution purposes, we observed reactions between these metabolites, and thus rather decided to turn to a more stable mixture of metabolites: glycine, sodium acetate and sodium formate.

The concentrations given are those of the impregnating solution (1 M for each species), and even in the case of a quantitative extraction they are expected to be diluted by the D_2O eluent to 12 mmol/L.

In practice, the extraction yield during dissolutions from HYPOP was found to be around 86 %.

Q8) I do not understand the ^{13}C spectrum given in Fig. 5a. The two signals on the left might correspond to acetate and formate respectively, but what about the peak around 160 ppm? Could the author provide an assignment?

R8. We added the assignment on the spectrum. The large ^{13}C - ^1H coupling split for formate is exacerbated by the low field of the spectrometer used.

Q9) Surprisingly, Glycine is absent on the ^{13}C spectrum. Could we imagine a specific adsorption of glycine on the polymer surface? More generally, do the authors consider that some category of molecules could be retained by the porous polymeric matrix?

R9. This is addressed in the response R11 of Rev1.

Until now, we have only observed an adsorption/diffusion behavior for ascorbic acid (See R7 of Rev. #1). As most metabolites being tested in the manuscript are charged at neutral pH, we believe that their solubility in the epoxy matrix is very poor.

Q10) The final radical concentration values given in Table S1 (17, 29, 44, 63, 95, 116, 191, 286 $\mu\text{mol.g}^{-1}$) are not consistent with the rest of the manuscript (16, 26, 45, 62, 93, 116, 192, 285 $\mu\text{mol.g}^{-1}$).

R10. We thank the reviewer for pointing out the mismatch in the table S1. We corrected it.

Q11) Overall, the quality of the SEM images is really poor, especially those at high magnification.

R11. We agree that the images are not of the highest quality despite numerous attempts to obtain better resolution. We believe that this is due to distinct effects:

First, the highly porous polymers materials may remain electrically insulating despite metallic coating, which may lead to local charge build-up and distortion of images.

Secondly, although the HYPOP are permanently crosslinked polymers, they are highly porous with T_g about 130 °C. We believe that local heat generated during the observation may be sufficient to deform the samples.

To alleviate both effects as much as possible, we worked with low voltage, which impacts the resolution and quality of the pictures.

Q12) In Fig S10, we do not really understand that is a “10 M 1H solution”. Moreover, the impregnated and Dry samples, that are compared, are not of the same amount of radical. R12. We changed to “10M of proton spin concentration” to be more precise and modified the text above : “Microwaves were optimized on one dry HYPOP, then compared with another impregnated HYPOP without observing any change on optimal frequency. We assumed similar pattern for all polymers (dry and impregnated) and thus kept 197630 MHz as microwave frequency for all DNP Build up.”

Reviewer #3:

This paper presents a new tempo-functionalized epoxy based polymer formulation as a polarising agent for dissolution DNP experiments. The formulations are shown to perform well for the polarization of model solutions (but at this stage not better than previous work, as the authors acknowledge), and the paper provides a detailed description of the optimisation of the polymer and the sample formulation.

The work well carried out and is a nice addition to the library of formulations for dissolution DNP, and the formulations introduced here have some features that are advantageous. However, the work builds on and combines the authors' own extensive previous work using either porous silica networks or microcrystalline formulations.

Response:

We thank referee #3 for his constructive criticism. We provide below answers to each issues that have been raised.

Q1. It is surprising that there is little to no mention of previous work using polymer networks as polarisation sources, which have been envisaged in a range of contexts for at least the last 10 years. An especially relevant early example is Phys. Chem. Chem. Phys., 2010,12, 5879-5882. Others include Macromolecules 2018, 51, 20, 8046–8053; Journal of Magnetic Resonance 190, 2008, 307-315; J. Am. Chem. Soc. 2018, 140, 22, 6969–6977; A quick search even yields an epoxy-tempo mixture <https://doi.org/10.1016/j.nima.2014.11.114>

R1. Thank you for bringing to our attention these articles, which indeed show that previous generations of polarizing media for DNP include a wide range of polymer with covalently bound Polarizing Agents (PAs).

An essential distinction should however be made between i) polarizing media for MAS-DNP, in which the hyperpolarization and NMR analysis is conducted on the same spectrometer and does not involve separation between polarization medium and hyperpolarized analytes, and ii) polarizing media for dissolution-DNP that must feature easily loading and extraction of polarized species and thus require optimized contact between the polarizing medium and target analytes.

We had indeed focused our article on d-DNP, but have added for the sake of clarity the references provided by the reviewer into a paragraph describing conventional DNP polarizing media.

Q2. The article "Producing Radical-Free Hyperpolarized Perfusion Agents for In Vivo Magnetic Resonance Using Spin-Labeled Thermoresponsive Hydrogel" from 2016, Macromol. Rapid Commun. 2016, 37, 1074–1078, which has a similar concept and objective to the present work, is mentioned as reference 33, but it is discounted in three words as "preventing hyperpolarization transport". In reality the work in reference 33 achieves the same separation between the PA and the analyte that is what (conceptually) enables transport here, but by a change in shape of the PA.

With this ensemble of background work in mind, the results here do not represent the kind of conceptual step forward I would expect from an article in Nature Communications.

R2. We would like to emphasize that while the article “Producing Radical-Free Hyperpolarized Perfusion Agents for In Vivo Magnetic Resonance Using Spin-Labeled Thermoresponsive Hydrogel” considered indeed a thermoresponsive hydrogel with LCST behavior, the phase separation between the PA and the analytes may only be effective above the critical temperature, i.e. after the dissolution, and not in the solid state in which storage performances are effectively measured (and remain relatively fast, about 40 min even at the temperature of 1 K).

Thus the objective of this reference and the underlying concept (which is effectively ingenious !) is not to achieve several hours lifetimes in the solid state to enable transport of hyperpolarization, but rather to produce radical free solution with liquid ^{13}C polarization lifetimes of a minute. We would like to emphasize that our new generation of HYPOP enables both features, which in our opinion represents a significant conceptual and technological progress.

We have rewritten the last part of the introduction to better emphasize the key concept behind HYPOP : increasing the effective separation between polarizing agents and hyperpolarized analytes through the use of macroporous materials.

As far as we know, such materials have never been used for dDNP applications, and HYPOP are the first materials to combine such extended solid-state relaxation time with the possibility to transport hyperpolarized samples and to obtain contaminant-free hyperpolarized solutions.

Q3. On a more minor note, the authors refer to polarization of "arbitrary" solutions. My reading is that only water:ethanol was used here, and that pure water was not suitable, and some organic solvents swelled the polymer. Only two quite similar samples are then given which are (i) 1M ^{13}C urea, 1M ^{13}C glycine, 1M ^{13}C sodium carbonate, 1M ^{13}C sodium formate and 150 mM of sodium ascorbate; and (ii) 0.9 M ^{13}C glycine, 0.9 M ^{13}C sodium formate, 0.9 M ^{13}C sodium acetate. This is anything but "arbitrary". The ascorbate is included as a radical scavenger, and the others are all chosen to have slowly relaxing ^{13}C centers. Of these the glycine, which is the only one with a non-exchangeable CH_2 group, is not observed! There is nothing wrong with this example, but it cannot be used to support the claim that arbitrary solutions can be polarized.

R3. We have indeed addressed the possibility to use pure water above and have indeed replaced the term “arbitrary solutions” by “mixtures of metabolites” (See R1 of Rev#2).

REVIEWERS' COMMENTS

Reviewer #1 (Remarks to the Author):

In their itemized responses to a number of my comments, the authors provide relatively general explanations without specific details or stating the locations where their changes have been implemented in the revised manuscript. This makes it difficult to find whether or how the authors have implemented several of their changes. I urge the authors to quote explicitly their changes and state at least the page numbers, and preferably the columns and lines, where their highlighted changes appear in the revised manuscript. Nevertheless, it appears that the authors have satisfactorily addressed my comments, criticisms, and suggestions. The extensive additions on pages 3-4 of the revised manuscript are particularly helpful and explain the context and significance of the present work. I recommend that the manuscript be accepted for publication.

Reviewer #2 (Remarks to the Author):

My questions and comments have been addressed accordingly. The paper is now suitable for publication in Nat. Comm.
I would have appreciated a version of the revised manuscript with changes highlighted. This would have avoided having to look for differences between the original and the revised version.

Reviewer #3 (Remarks to the Author):

The authors have addressed the numerous question raised by all three reviewers in a comprehensive rebuttal, that involves re-wording the manuscript. No extra experiments or analysis have been added.

Regarding my comment that the progress made here is detailed and incremental, especially in the light of the articles that they had not previously cited, therefore nothing has changed. I maintain my original opinion.